# A Review of Sensing Technologies for Non-Destructive Evaluation of Structural Composite Materials

**Ranjeetkumar Gupta [1,2,\*]**, **Daniel Mitchell [2]**, **Jamie Blanche [2]**, **Sam Harper [2]**, **Wenshuo Tang [2]**, **Ketan Pancholi [3]**, **Lee Baines [4]**, **David G. Bucknall [1]** and **David Flynn [2]**

[1] Materials Group, Institute of Chemical Sciences, School of Engineering and Physical Sciences, Heriot-Watt University, Edinburgh EH14 4AS, UK; david.bucknall@hw.ac.uk
[2] Smart Systems Group, Institute of Sensors, Signals and Systems, School of Engineering & Physical Sciences, Heriot-Watt University, Edinburgh EH14 4AS, UK; dm68@hw.ac.uk (D.M.); j.blanche@hw.ac.uk (J.B.); sh324@hw.ac.uk (S.H.); wenshuo.tang1@hw.ac.uk (W.T.); D.Flynn@hw.ac.uk (D.F.)
[3] School of Engineering, Robert Gordon University, Aberdeen AB10 7GJ, UK; k.pancholi2@rgu.ac.uk
[4] MacTaggart, Scott and Company Limited, Loanhead, Midlothian EH20 9SP, UK; lee.baines@mactag.com
\* Correspondence: r.gupta@hw.ac.uk; Tel.: +44-131-451-4553

**Abstract:** The growing demand and diversity in the application of industrial composites and the current inability of present non-destructive evaluation (NDE) methods to perform detailed inspection of these composites has motivated this comprehensive review of sensing technologies. NDE has the potential to be a versatile tool for maintaining composite structures deployed in hazardous and inaccessible areas, such as offshore wind farms and nuclear power plants. Therefore, the future composite solutions need to take into consideration the niche requirements of these high-value/critical applications. Composite materials are intrinsically complex due to their anisotropic and non-homogeneous characteristics. This presents a significant challenge for evaluation and the associated data analysis for NDEs. For example, the quality assurance, certification of composite structures, and early detection of the failure is complex due to the variability and tolerances involved in the composite manufacturing. Adapting existing NDE methods to detect and locate the defects at multiple length scales in the complex materials represents a significant challenge, resulting in a delayed and incorrect diagnosis of the structural health. This paper presents a comprehensive review of the NDE techniques, that includes a detailed discussion of their working principles, setup, advantages, limitations, and usage level for the structural composites. A comparison between these techniques is also presented, providing an insight into the future trends for composites' prognostic and health management (PHM). Current research trends show the emergence of the non-contact-type NDE (including digital image correlation, infrared tomography, as well as disruptive frequency-modulated continuous wave techniques) for structural composites, and the reasons for their choice over the most popular contact-type (ultrasonic, acoustic, and piezoelectric testing) NDE methods is also discussed. The analysis of this new sensing modality for composites' is presented within the context of the state-of-the-art and projected future requirements.

**Keywords:** non-destructive testing (NDT); prognostic and health management (PHM); eddy current (EC); shearography; infrared thermography (IT); computed tomography (CT); ultrasonic testing (UT); acoustic emission (AE); digital image correlation (DIC); frequency-modulated continuous wave (FMCW)

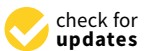

## 1. Introduction

Composite structures have a broad range of applications due to their low weight to modulus and strength ratio, cost-effectiveness, and efficacy. The proven high demand for the load-bearing composite structures in industrial sectors includes onshore and offshore renewables [1], automotive [2], civil infrastructures [3], and aerospace [4]. The extensive involvement of composite materials in these application areas is summarised in Figure 1.

The effective utilisation of composites in a wide range of these applications requires non-destructive evaluation (NDE), which is executed during selected intervals and for continuous structural health monitoring (SHM), or its advanced version as prognostics and health monitoring (PHM) that also allows for forecasting and avoiding any sudden critical failure, as part of the manufacturing process and during their in-service lifetime. Even composites' production involves multiple variables over its production cycle, and the likelihood of occurrence of different defects is high and increases the significance of safety concerns during the service life [5,6]. This amplifies the need for NDE during the manufacturing process as well.

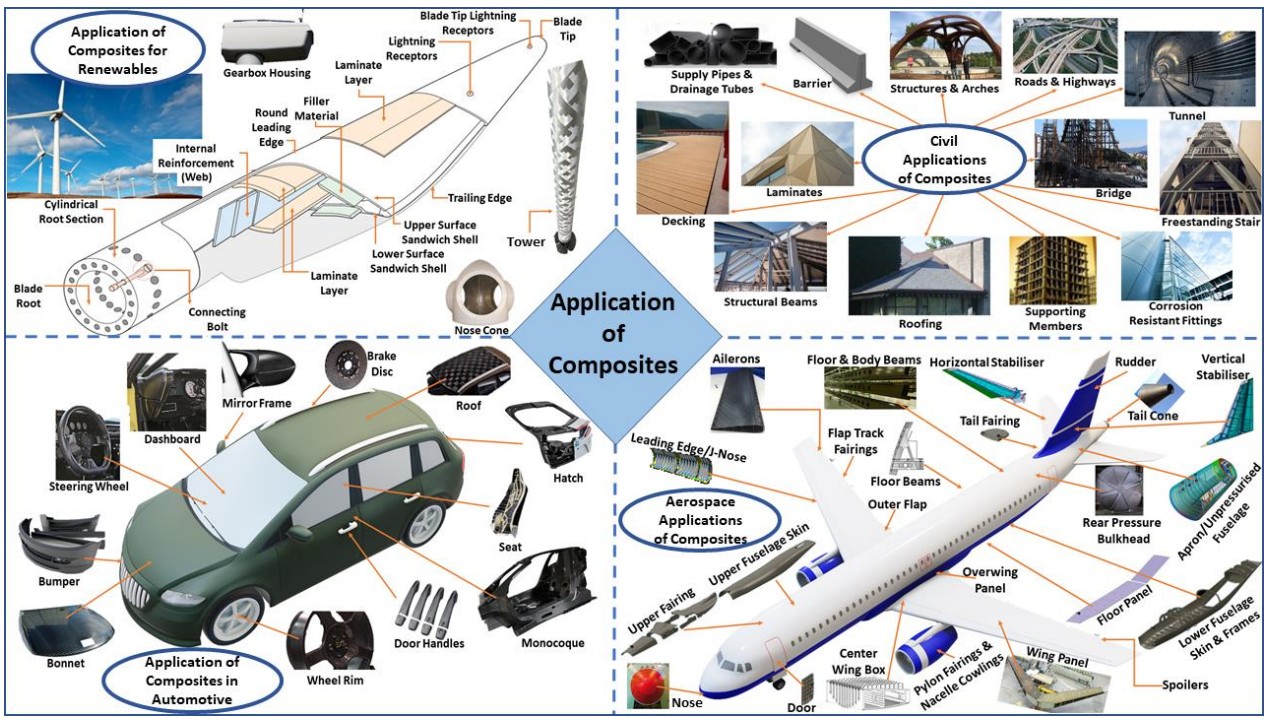

**Figure 1.** Application of composites in the areas of renewables, civil infrastructures, aerospace, and automotive (clockwise from top left).

At the composite structures' manufacturing stage, the cost can escalate due to a variety of factors, including parameters such as slow curing cycle time and various other variables contributing to the failure of achieving the reproducibility across all batches. Some of the process parameters, such as filling and curing, affect the quality of the final product. Filling defects, such as inclusions and porosity, or cure-induced defects, such as residual stresses and degraded under-cured product, represent a risk to achieving the required structural properties of the final composite and need be eliminated at the quality control stage. Additionally, the process inputs such as raw material, geometry, or relevant thermodynamic conditions are also important for quality control at the process stage as these parameters dominate the mechanisms of filling and curing in the manufacturing of composites [7].

Maintaining structural integrity during operation and manufacturing requires meeting the predominant challenges to reduce the intrinsic noise during the detection and evaluation of the defects within the anisotropic and non-homogenous composite structures [8]. The occurrence of defects and damage within the structure can be at any location and at different scale length levels, hence tracking the defect sites can become difficult and remain undetected for a long time, leading to accumulation of the damage. Finally, the damage accumulation in composite structures is associated with the reduction in the stiffness, strength, and service life of the components [6,9]. Figure 2 presents an overview of the categorisation of various defects based on the scale level.

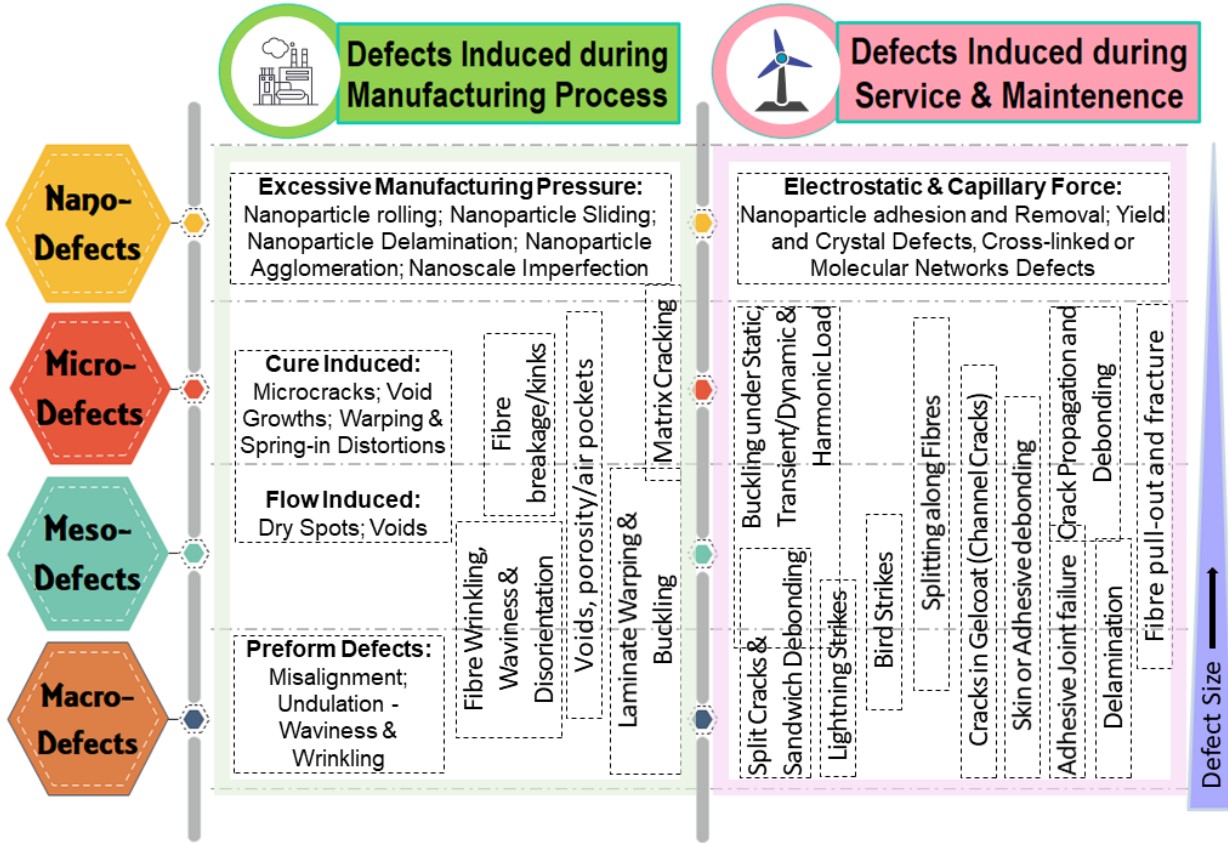

**Figure 2.** Evolution of defects induced while manufacturing and during in-service of composite structures, with the defect scale dimension included.

Engineers and scientists have a particular interest in structural health monitoring as it allows more precise and in-depth prognostics for enhanced asset management and data-driven design of future composite structures. As this allows for fast and accurate damage evaluation of composites, this precise monitoring has applications in various fields, such as space orbital delivery systems and satellites, offshore renewables [1], and in the military and civil aerospace sectors [4]. The accurate information of the current stress state and the accumulated damages which have occurred in the structure or component is combined with statistical analysis of the defect index for the component, where the current and cyclic loadings help in determining the remaining useful life (RUL) [10], which is equivalent to the conventional safety factor for the part. This can result in cost-saving improvements for the maintenance of wind turbines, aircraft, etc., and also reduce the duration of repairs, when compared to the current techniques of visual inspections, to assess the structural integrity of components [11].

Structural health monitoring can be categorised into two methodologies: passive and active sensing. The difference between passive and active sensing is that in the former, no form of energy is imparted into the test specimen with sensors just deployed in detection mode for gathering structural health data. Passive structural health monitoring is where signals are recorded from sensors continuously, such as in optical fibre sensors [12] to identify the structural condition from analysing the recorded signals [13]. Passive sensing is used to identify the unknown inputs, such as incipient flaw location, which can result in detectable variations in sensor readings. Active sensing methods can be used as a transmitter and receiver response, for example piezoelectric transducers, to examine the structure with definite excitations and record the equivalent responses. Therefore, active structural health monitoring uses known inputs and relates variations in local sensor measurement for the identification of flaws and can be used over potentially large surface areas, such as aircraft wing and fuselage structures [14].

Given the complexity of composites, precursors to failure can be highly localised, and therefore external systems can face difficulty in detecting and locating the point of origin. To facilitate the detection, the sensors can be deployed as an integral part of the structure to read the stress and strain at the location. Unlike isotropic materials, the lamination architecture of the composite facilitates the embedment of the sensor within the structure and aids in risk reduction, wherein the suite of inbuilt sensors may be tested and characterised prior to asset installation [15]. These embedded sensors can suffer from drift and failure events over their lifetime and data transmitted from such sensors can wrongly be inferred as an "event"—a precursor to a failure [16]. Challenges relating to the effective implementation of prognostic and health monitoring (PHM) systems are also associated with the need for expert elicitation of the data, the complex data analysis algorithm, and scalability of such algorithms, for example, adapting data to intrinsic variances in materials used on the structural assets [17]. In terms of deployed, external monitoring solutions, these can suffer from issues of asset coverage (scanning rate) and accessibility to assets at height or located in hazardous environments [18]. Another issue associated with the interpreting process of collated data is the interpretation of stress and strain data at the defect location in addition to detected defect accuracy [11]. There are also challenges associated with sensor system deployment due to the need for lightweight systems that do not obstruct the structural integrity of components (assets) within the sensors' implementation [15]. Some external monitoring technologies also only provide a surface analysis of these structures, with some also requiring a stimulus, e.g., thermography, applied to the asset which can exacerbate precursors to failure.

Despite the advances in sensor approaches to monitor composites, the most reliable and detailed methods for monitoring are NDE of composite structures, which is important for minimising safety issues and maintenance costs to reduce the likelihood of process downtime and interruptions to normal operations [7]. NDE involves a large variety of methods for different types of defects' detection in composite material or structural characterisation [19]. NDE methods have been exploited for assessment and evaluation of composites utilised in a variety of industrial sectors, such as storage tanks, pipes, and tubes [20–22], aircraft, UAVs, maritime vessels, land weaponry systems, and defence weapons [23], window frames, floor beams, fuselage panels, pressure bulkheads, fittings, etc., in aerospace vehicles [24], onshore and offshore composite wind turbine (WT) blades [25], and structures used in the nuclear industry [26]. There are various NDE methods that are popularly used for composites, which include visual inspection [27], liquid penetrant testing [28], ultrasonic testing [29], thermographic testing [30], acoustic emission testing [31], acousto-ultrasonic testing [32], optical testing [33], infrared thermography testing [34], etc. There has been an emergence of the non-contact-type NDE method, such as digital image correlation and the disruptive frequency-modulated continuous wave (FMCW) method [35–37], as the composites' NDE with the contact-type methods (e.g., acoustic and ultrasonic methods) have shown concerns of accessibility and the effectiveness of the method itself.

Considering the increasing demand for NDE of composites to support the sustainable development plans in various application areas, for example, renewable energy development supported by composites, it is important to have an in-depth understanding of the working principle, benefits, and limitations of the state-of-the-art NDE methods. Though there have been multiple review papers published on this topic over the years [6,38], and recently also focusing on a particular sector [39] or method [40], a recent review document encompassing the general principle of popularly used NDE methods and their progress to date is identified as a research need. This review paper discusses the working principle for structural composites in general, that can be extended to specific application areas. This review paper firstly sets out the background for the need and importance of NDE of composites, and in Section 2 indicates the progressing trends for these techniques. A summary of the most suitable and applicable composite NDE methods are presented and categorised based on their types in Section 3, and a discussion is included on the work-

ing concepts, limitations, and benefits of each NDE method with respect to their use for structural composites. In Section 4, a comparison is made of all the NDE methods in a tabulated form to provide the reader with a quick understanding for choosing a suitable NDE method for a required application at hand. Lastly, the emerging trends and future of NDE of composites is discussed in Section 5 with the conclusions following based on the overall discussion.

## 2. Progressing Trends in NDE of Composites

The growth in demand of various composite NDE techniques can clearly be gauged from the increasing number of research work published over the past 30 years, as shown in Figure 3, where the trend suggests significant continued growth.

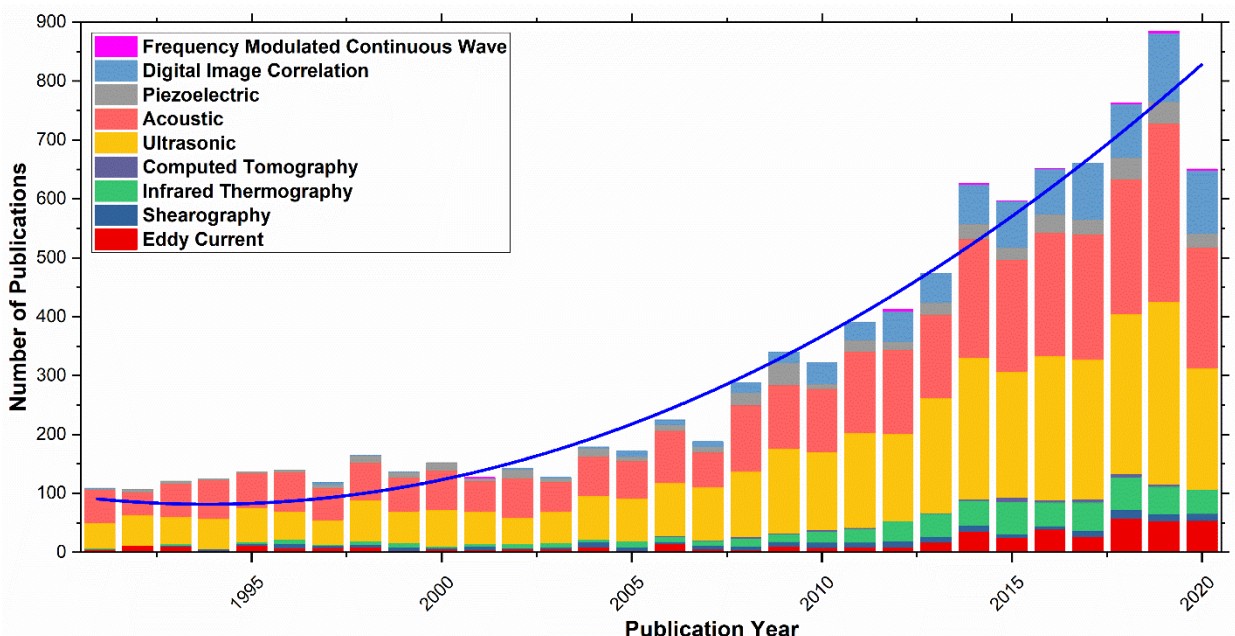

**Figure 3.** Composites NDE trend based on the published research on various methods for the last 30 years. The data were generated using the SCOPUS database with a keyword search of composite structures and materials' NDE. The keyword search results were categorised based on type of NDE method and the corresponding year of publication. The solid line is a trend line for the total number of papers.

In a market report published by Mordor Intelligence in 2020 [41], the NDE market was valued at USD 16.72 billion in 2019, with predictions for a 6.7% CAGR to 2025, giving an estimated market value of USD 24.65 billion [42]. The British Institute of Non-Destructive Testing states that in the UK, more than 25,000 daily inspections are conducted on-site and in factories to identify flaws and defects across all manufacturing, the majority of which are associated with composite components. A key driver in NDE market growth is the requirement to evaluate the conditions of an aging asset base. This driver is facilitated by the increasing rate of adoption of Internet of Things (IoT) devices.

The demand for flaw detection associated with porosity, cracks, or manufacturing disorder has substantially increased with the development of automation in industrial and infrastructural areas. Thus, requirements for industrial safety standards and structure reliability certifications are driving the growth of the NDE market. Governmental agencies such as the International Organisation for Standardisation (ISO) and the American Society of Mechanical Engineers (ASME) are global institutions tasked with ensuring procedures and regulations are followed by engineers, assuring safety sensing instruments and supervision for engineering testing services, all factors that positively influence the NDE market worldwide [41].

### 3. State-of-the-Art Review of NDE Methods

NDE methods utilise variable portions of the frequency spectrum to perform characterisation of defects and flaws. The choice of a particular range of the frequency spectrum to utilise in any particular NDE application depends on a number of factors, including penetration, resolution, and contrast. Based on the operating frequency and the technique involved, different types of NDE methods can be categorised into imaging technique-based, chemical spectroscopy-based, electromagnetic spectrum-based, and acoustic wave-based (see Figure 4).

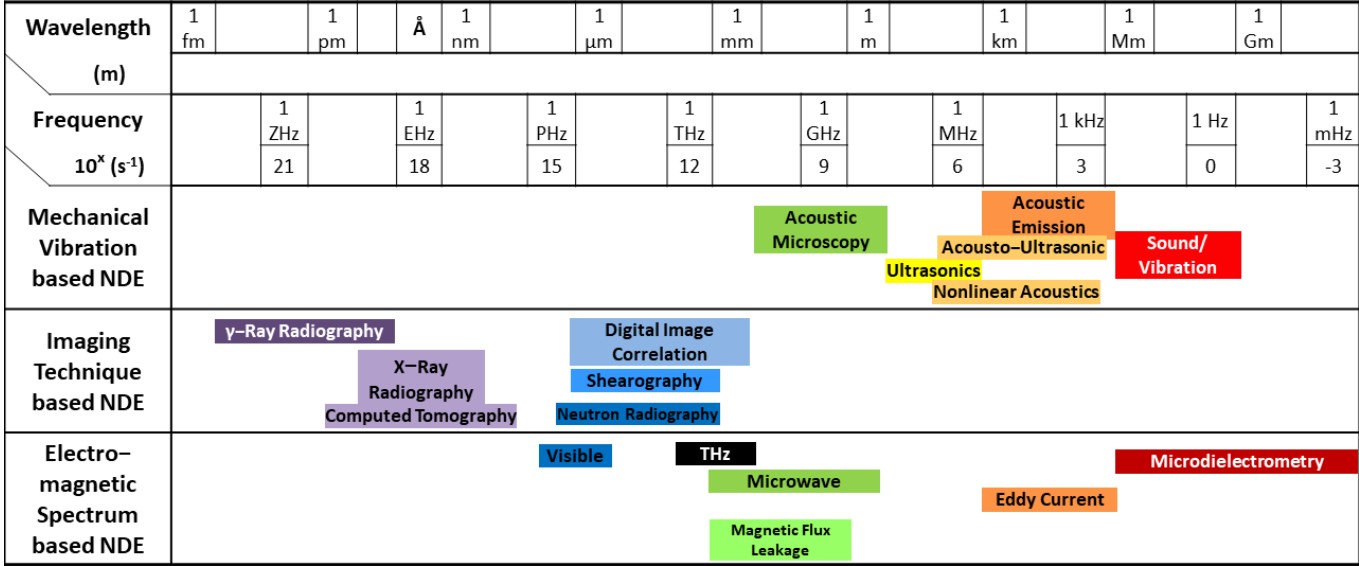

**Figure 4.** Different types of NDE methods mapped to their frequency domain.

The popular NDE methods under each category are discussed as follows.

#### 3.1. Mechanical Vibration-Based NDE

This method involves different modes of emitted energy emissions which propagate into the solid [43]. The most popular techniques used in this type of analysis are electrostatic transducer-based ultrasonic testing, piezoelectric transducer-based ultrasonic testing, and acoustic emissions testing. These methods are discussed briefly below.

#### 3.1.1. Electrostatic Transducer-Based Ultrasonic NDE

This ultrasonic testing method comprises electrostatic transducer tools, that act as a transmitter and receiver unit separately, and a display device. The information achieved from the signals is based on defect size, orientation, crack location, and other features. The working principle is depicted in Figure 5, wherein the transducer mounted on the specimen receives the signal associated with the internal flaw and displays its location on the monitoring detector unit. The application of this technology is in assembly line testing, wherein copies of design parts must be tested frequently [44]. Ultrasonic NDE has two types which are generally used for various applications: "pulse echo" and "through transmission" approaches. The ultrasonic testing of these two types uses sound waves with higher frequency in the order of 1 to 50 MHz to identify inner defects pre-set in the system [38]. Ultrasonic testing is carried out in three different modes: back scattering, reflection, and transmission, all of which use a transducer, a range of frequencies, and a coupling agent [45].

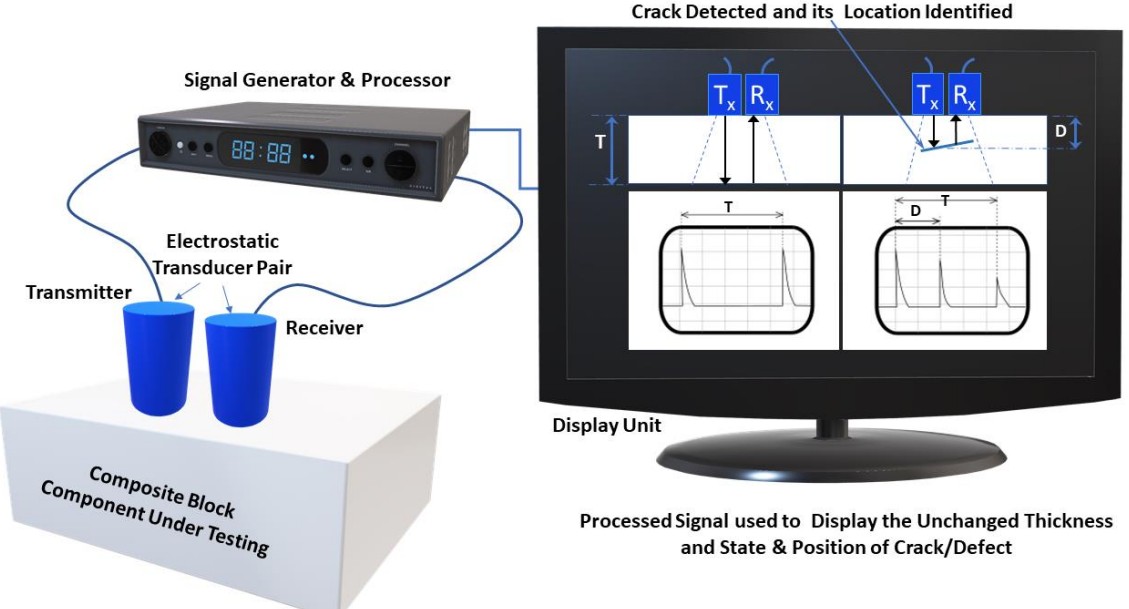

**Figure 5.** The working setup and principle of electrostatic transducer-based ultrasonic NDE of composites (Labels: $T_x$—Transmitter, $R_x$—Receiver, T—Thickness of material, and D—Depth at which defect is detected).

The pulse echo ultrasonic technique can freely identify flaws in homogeneous material. In this technique, the concerns of the operator are about the wave transit time and the loss of energy due to the wave scattering and attenuation on defects. The recorded variations in the wave propagation assist in locating the irregularity in the material [46]. The velocity measurements of ultrasonic pulses result in the detection of defect locations, large (~5 mm width) defects, quality control, and imaging purposes [47].

The "through transmission" ultrasonic technique differs from the conventional technique as the transducer and receiver have a non-contact configuration and are maintained at a set distance from the material. This method is particularly valuable when the intricate geometries are unable to contact the conventional transducers and receivers to the surface of the sample. The wave propagation velocity and energy or amplitude loss are the most frequently used indicators of properties [48].

Ultrasonic testing has advantages of flaw detecting capabilities, good resolution, short scan time, and is portable enough to be deployed in the field. The disadvantages of this technology include: complicated to setup, required accurate part scanning skills, and the desired test specimen requirements (low effectiveness in thin materials, relatively smooth surface is needed to couple transducer) to assure precise examination. There are still some limitations encountered in ultrasonic testing while detecting discontinuities in non-homogenous materials due to multi-reflections and high wave scattering, for example, in sandwich panels and composite laminates. This technique is not suitable for composite flaws at depths greater than ~50 cm from the material interface, where lesser resolution is attained when compared to the same thickness of steel. This is due to ultrasonic wave attenuation, which is sourced from absorption in porous resin and fibre scattering [49]. For that reason, it requires the use of lower ultrasonic frequencies in composite testing to decrease the attenuation coefficient in comparison with the homogenous materials [50]. Therefore, the depth of penetration is decreased so that ultrasonic testing is often not suitable for the characterisation of defects which exist far below the surface of the composite structure. In addition, the limited capability of electrostatic transducers affords the overall capacity of defect detection within 1–5 mm [51]. Additionally, the attenuation coefficient in ultrasonic waves can also be influenced by the shape, size, and spatial distribution of voids in composite materials. As a result, significant measurement errors of order $\pm 25\%$ are typically observed [52,53]. The other limitation of ultrasonic testing is encountered

while testing aerospace composites, which is defined as the shadow effect. The cause of the shadow effect is any delamination or large defect which is present near the surface. These large defects reflect most of the ultrasonic energy and result in low visibility below the discontinuity, hence resulting in a shadow [54].

### 3.1.2. Piezoelectric Transducer-Based Ultrasonic NDE

This ultrasonic technique is useful in inspection of non-porous and homogeneous materials [55]. Additionally, it is highly efficient in the inspection of laminated structures during quality control of delaminated areas, because of the capability of ultrasonic waves to be concentrated in small regions. The piezoelectric transducer is considered an important tool for these inspections, that can operate either as a source or a detector of ultrasound signals simultaneously, which makes it very popular compared to the electrostatic transducer-based ultrasonic measurements. The ability of performing both functions equally is because of the reversibility of piezoelectric effects and the independent reflection and transmission constraints concerning the direction of the working defects [55–62]. Herein, the commonly used ultrasound techniques are through transmission and pulse-echo techniques [55,56]. The through transmission technique involves two piezoelectric transducers, with one acting as a receiver and one as an emitter, mounted on opposite sides of the samples. When the ultrasonic signals reach the defects, they will be partly reflected and received by the transmitter, wherein a reduced signal is received by the receiver. The internal defect examination is performed by the proportion of these two signals. The pulse-echo technique requires one transducer, which serves both as a transmitter and receiver of reflected signals [57]. The simplest construction of a piezoelectric transducer is shown in Figure 6 and consists of a cylindrical shaped piezoceramic element, which is aligned normal to the parallel faces in the single-direction axis of polarisation.

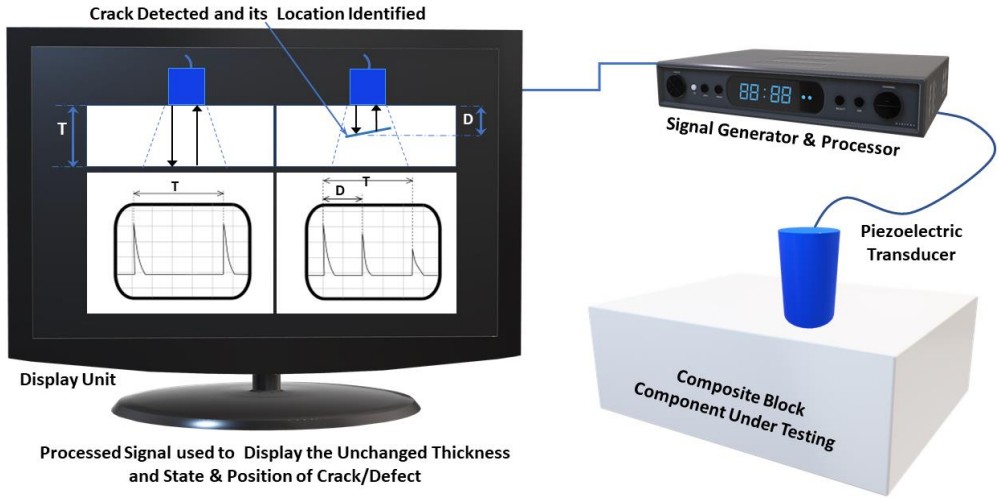

**Figure 6.** The working setup and principle of piezoelectric transducer-based ultrasonic NDE of composites (Labels: T—Thickness of material and D—Depth at which defect is detected).

Piezoelectric transducers operate via high-frequency ultrasonic vibrations generated by reactions to short electrical pulses in the piezoceramic element. Conversely, reflected high-frequency sound signals are received, which are then transformed into electrical signals. The generated vibrations of the end face of this piezoceramic element are similar to piston-like motions, with sufficient directivity and fine separation of harmonics if the cylinder diameter is adequately large, in comparison with thickness. Conversely, large thickness would be required if the frequencies required are lower so that the cylinder or disc becomes inconveniently large. Sandwich-type piezo active elements can be used for this purpose, as these elements are made from several thinner discs, stick together, and

have opposite polarisation directions, and are then tightly attached on a steel cylinder playing the function of a backing material.

Piezocomposites have many advantages for NDE applications due to their flexibility and wide bandwidth. It is useful to have high flexibility of piezocomposites in inspecting curved composite pipelines or steam pipes for monitoring pipe curvatures, and this can result in wave loss, significantly decreasing the signal-to-noise ratio. The piezoelectric transducers have better resolution than electrostatic transducers and hence can be used for identification of critical defects of relatively large (up to 25 mm) size [27]. However, there are also key challenges of piezoelectric material in the operation of NDE methods, of which the most important is the material survival in a high-temperature application. The applications of traditional piezocomposites are being limited in high-temperature and high-power applications because of the naturally high thermal expansion of polymer fillers, low mechanical quality factors, and low thermal conductivity. The high thermal expansion coefficient of polymer fillers can cause debonding and cracking in the composite structure itself, resulting in structural failure at high temperatures. Low thermal conductivity of polymer can decrease the thermal dissipation to neighbouring environments and cause a localised hot spot near the piezoelectric pillar in composites, which results in polymer melting. Low mechanical quality factors can result in internal heating and power loss when working under high powers at resonance [63–65].

### 3.1.3. Acoustic Emission-Based NDE

In this method, the mechanical vibrations are generated by defects encountered in the material: localised delamination, fibre breakage and pull-out, matrix micro-cracking, or matrix and fibre debonding [66,67]. These types of defects result in stress waves, which spread out from their sources concentrically and are identified by a tremendously sensitive piezoelectric array. Acoustic emission (AE) is effective for the imperfection analysis in composite materials or structures. The schematic arrangement for this method is shown in Figure 7. This technique is different in two characteristics from most other NDE methods. The first feature is the source of the signals. This technique is centred on the release of sound energy within the material under test, instead of supplying energy to the material. The ability to distinguish the development of dynamic faults, in addition to inactive, non-critical defects, is its key impact [38].

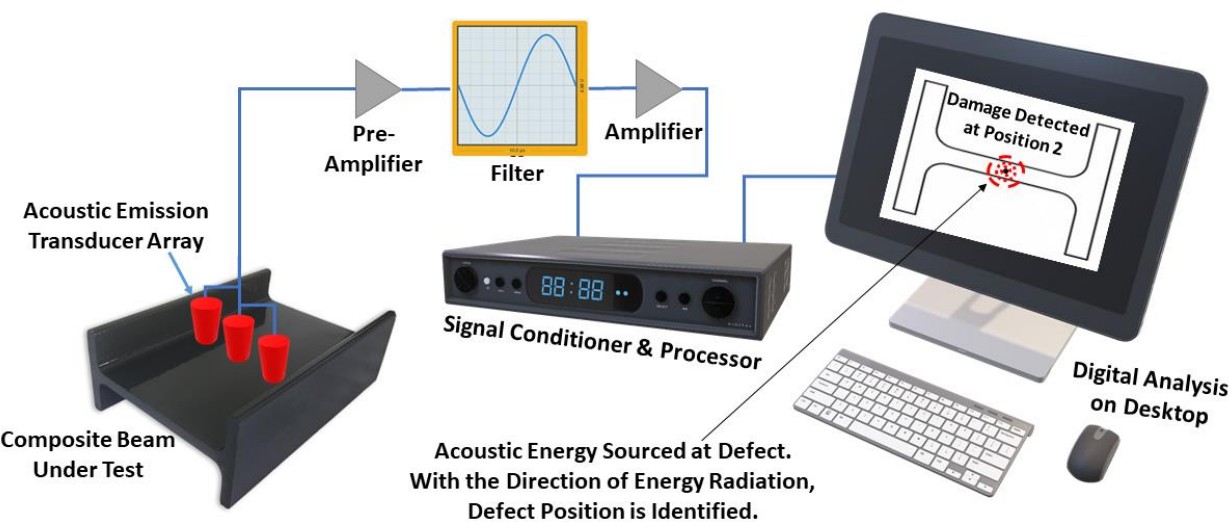

**Figure 7.** The working setup and principle of acoustic emission testing-based NDE of composites.

Every AE event represents a discrete stress wave which can neither be stopped nor reproduced. It implies that the nature of the signal source cannot be reproduced by any particular test. For example, slower crack growth produces weak acoustic emission signals, whereas fast crack growth with the same source size generates transient signals [68].

Furthermore, the AE signals in composite structures will acquire considerable changes when travelling through the transmission pathway, and they are recorded by the receiving AE sensors. AE sensor coupling is also necessary, and the procedures have little descriptive value with regards to discontinuity in the data [14].

AE has various other advantages of quick and overall testing by means of multiple sensors, high sensitivity, sensor mounting for process control, and no requirement of disassembling and sample cleaning [69]. In addition, the technique is helpful in detecting various defect types resulting from fatigue loading. The fatigue damage-type defects detected by AE testing are fibre/matrix debonding, fatigue cracks, matrix micro-cracks, delamination, and fibre fractures. Only the defects from sub-millimetre to up to 5 mm can be effectively studied [67]. The negative aspect of AE testing is that it requires high skill for correlating collected data to an explicit damage mechanism type [38].

### 3.2. Imaging Technique-Based NDE

The imaging technique-based NDE identifies the difference in the captured images before and after a defined time/deformation, which highlights the changes due to a flaw or defect. Some of the popular imaging NDE techniques are shearography testing, computed tomography, and digital image correlation, etc. The principles of these popular imaging technique-based NDE methods are discussed in this section.

### 3.2.1. Shearography-Based NDE

Shearography is a laser-based method, the basic layout of which is shown in Figure 8. A laser source is used to illuminate the sample, which is imaged with the charge-coupled device (CCD) camera via the beam shearing element. The laterally shifted subsequent images of the sample surface that are continuously captured are coherently superimposed in the image plane by the optical beam shearing element. This captured lateral shift is termed as image shear and the created superposition is called a shearogram. The shearogram is an interferogram created over the reference object wave with the superimposition of a sheared object wave over it. With variable loading conditions, multiple images for shearograms are similarly recorded, wherein the induced deformation or variations are captured. The difference in the deformation state due to loading variations is then correlated with the interference fringe pattern resulting from the absolute difference recorded in subsequent shearograms. This resulting differential image is further termed as a "D-Image". When processed, rather than providing deformation (as in holographic interferometry), the fringes provide the rate of change of the deformation. The surface and subsurface defects tend to modify due to the applied loading, resulting in minor alterations or major disturbances in the recorded loading fringe pattern, which is expected to appear more or less uniform for the no defect case. Hence, this principle is used for classifying and categorising various defects, depending on the extent of alterations or disturbances recorded in the shearographic fringe pattern. The simplified working principle of shearography testing is shown in Figure 6. Although, it is essential to induce deformation in the sample as applied by vibration [70], mechanical loading [71], thermal expansion or contraction [72,73], vacuum force [74,75], and microwave heating [76], and it could be applied in a static or dynamic way. The CCD camera captures the interferometric pattern, which leads to an edging image and consists of structural information [6,77].

A loading system is used to stimulate deformation or to change the state of deformation of the sample surface, which is required in shearography testing. The loading systems which are normally used in shearography comprise of thermal pulse shearography, vibration shearography, and vacuum shearography. Thermal pulse shearography is efficient for inspecting impact damages or cracks that are barely visible to the naked eye. When the image shearing direction is not perpendicular to the orientations of cracks, the detected defect direction has sensitivity which is comparatively greater than the perpendicular image shearing [72]. Vibration shearography is effectively utilised for the inspection of foam on the external tank of NASA's space shuttle [78], and also to disclose flat-bottom

holes of variant sizes and locations at various depths in composite laminate [70]. Vacuum shearography is efficient for imaging fibre debonding in composite laminate [79], aluminium honeycomb panel [80], and also in the composite element panel of the helicopter (honeycomb core with two outer layers of epoxy and graphite) and in the tail unit [81] for core damages, core splice-joint separation, and delamination [78]. There are other popular methods, such as thermoelastic stress analysis, that involves an infrared camera (replacing the laser source and CCD camera, as shown in Figure 8) for picking any variations, which is very similar in principle with shearography [82,83] and widely used for continuous monitoring of composites.

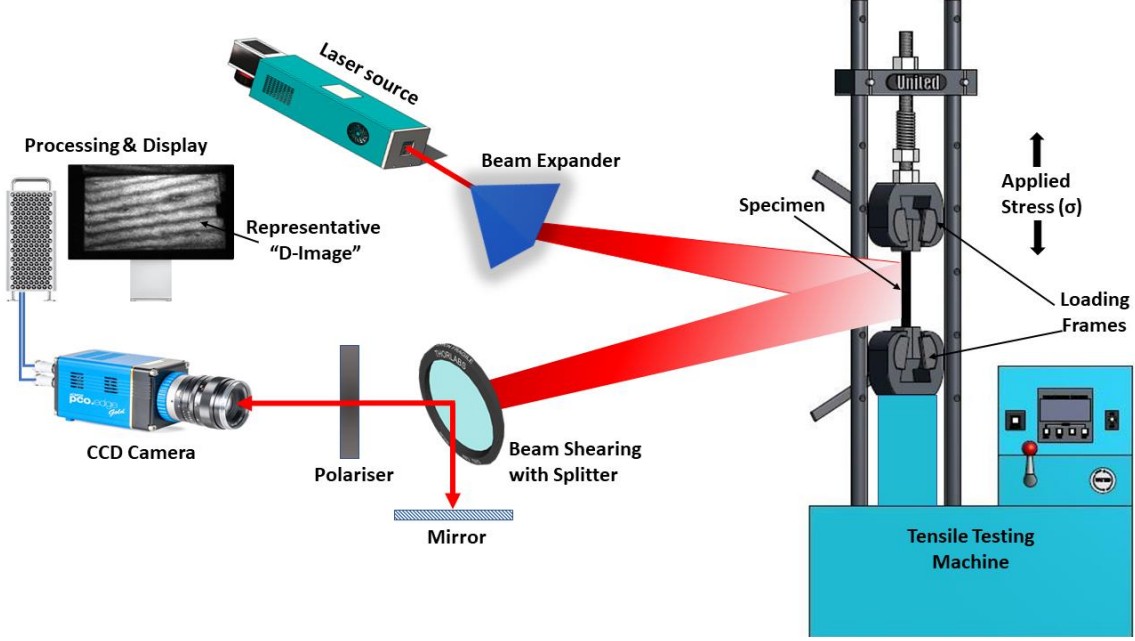

**Figure 8.** The working setup and principle of shearography testing-based NDE of composite components.

The advantage of shearography includes that by easily highlighting stress concentrations around the specific defect, it highlights the type and criticality of that defect, and since composite failure normally occurs by stress concentration, the degree of stress makes a lot of difference [84]. The other advantage of shearography includes it being less prone to noise than other different types of NDE. This feature is useful because it does not require highly skilled operators for the inspection and determination of component usability without long-term training, since just comparing the deformed and undeformed shearograms becomes a lot easier. It has been found as very useful method for honeycomb and foam composite structures, with the ability to detect defects up to 2–3 mm in depth or sometimes even more [27]. The major drawback associated with shearography is the difficulty of inspecting defects other than delamination. For that reason, it is sometimes combined with other NDE types which can help in identifying specific flaws [38].

Another noteworthy shearography limitation is the requirement of applying appropriate external loading increments to the testing sample during examination. Therefore, appropriate loading systems are required. Furthermore, the alteration monitored in the displacement pattern derivative reduces with the defect depth or with an increase in its diameter. Therefore, the efficient digital shearography application for defects characterisation is difficult and is still dependent upon various factors, for example depth and defect type, material type, and laser illumination [85]. Hence, this is another reason why shearography is sometimes coupled with other techniques of testing to detect flaws other than delamination [38].

Overall, there are two major drawbacks associated with shearography. The first is the difficulty of detecting defects other than delamination. For this reason, it is sometimes

combined with other NDE techniques to help in identify other types of flaws [38]. Secondly, shearography requires applying external loading to the testing sample during examination, and consequently, an appropriate loading system is required. Furthermore, the alteration monitored in the derivative of the displacement pattern reduces with the subsurface depth of the defect or with the increase in its diameter. Consequently, application of shearography for defects' characterisation is often challenging as it is dependent upon various factors, including the depth and type of defect (delamination, impact, crack, etc.), material type, and laser intensity [85].

### 3.2.2. Computed Tomography-Based NDE

Computed tomography (CT) is an advanced form of conventional X-ray radiography, which is used for non-destructive 3D imaging of internal features of solids. The working setup of CT-based NDE is schematically presented in Figure 9. This is an outstanding imaging technique to examine the details in terms of size and volume of structures with very high precision, and also in three dimensions, which is especially valuable for the inspection of structural integrity of complex geometries [86]. The resolution of the technique depends inversely on the volume measured. Consequently, standard CT, microCT, and nanoCT techniques have been developed for increasing the resolution of feature sizes at the cost of the 3D volume that can be imaged.

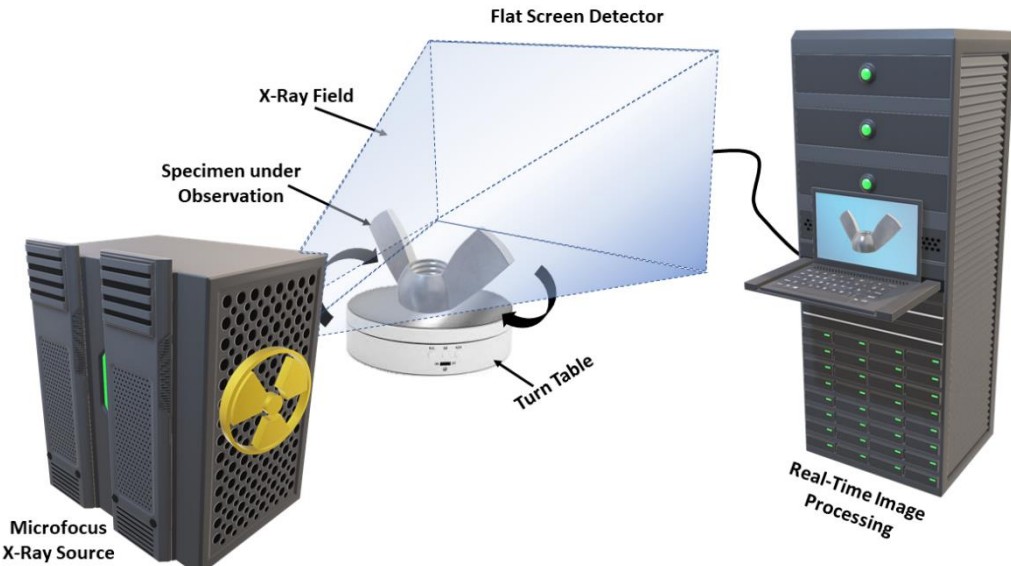

**Figure 9.** Working setup and process of computed tomography-based NDE of composite components.

The extraction of information from a computational tomography dataset involves a series of steps. The data are acquired from multiple radiographs obtained as the sample is rotated relative to the X-ray source. A reconstruction algorithm combines all the angle-dependent radiographs into a 3D reconstructed image of the sample [87,88]. Most computational tomography systems apply a filtered back projection (FBP) reconstruction algorithm, because of its predictable nature with regards to reconstruction times and computational cost [88]. The accurate representation of an object using FBP can be attained by projecting the X-ray integrals for each X-ray path back through the object. This projection method has high accuracy with the additional feature of low noise of the projected images, although alternative iterative reconstruction techniques have considerable advantages in more problematic settings [89,90]. Iterative algorithms utilise a linearised forward model of the X-ray acquisition method and use optimisation algorithms to reverse this model. Image viewing and processing techniques can be used to extract valuable information once a computational tomography volume is reconstructed, and this is called visualisation. The extremely high resolution achieved in nanoCT scans can detect details up to 0.2 μm for low

absorbing materials [91]. The obtained image quality is generally dictated by the variable control of spatial resolution [92], contrast, noise, and artificial features, called artifacts [92], such as scatter and beam hardening [93–95].

The limitation associated with computational tomography is the sample size, which greatly affects the details obtained by the current generation of computational tomography systems [96]. The resolution is restricted by the pixel size of the detector, which depends on the component geometry and is often 2 to 3 times the pixel size [97]. The affected region which the detector covers is normally 2000–4000 pixels wide [98], and therefore, the test object size is restricted by it. Other shortcomings are field of view limitation, in situ monitoring, and attenuation contrast. Comparing standard CT to microCT and nanoCT systems, the pixel size limitations indicated above can be translated into dimensions, i.e., millimetres, micrometres, or nanometres, which are more tangible and relevant than pixel size.

### 3.2.3. Digital Image Correlation-Based NDE

Digital image correlation (DIC) is a non-contact method to examine defects and has applications in structural composites. The sensing mechanism can perform inspections on active and passive structures. It is an optical technique which uses pattern matching and image registration methods for exact two- and three-dimensional calculations of change in the object shape which is being inspected [99–101]. The DIC technique is useful to determine deformation, stress, strain, and displacement. This method has a number of applications in engineering and manufacturing techniques to determine the changes and provide measurements for finite element analysis, material and structural analysis, and quality control [102–104].

The three-dimensional digital image correlation works on the principle of combined methods of image correlation with the photogrammetric location. Photogrammetry works on the triangulation principle, which is used for three-dimensional coordinate measurements [102], as shown in Figure 10.

Objects being examined are targeted in photogrammetry and a series of photographs are taken from different angles for recreation of dimensional target locations of the object. The accurate location of every target can be acquired by triangulation with various different target views of the object being examined [102]. Prior knowledge of the orientation and position of cameras for the images taken is important, and triangulation is dependent on these factors in photogrammetry. There are two cameras in 3D DIC, mounted at each end of a tripod camera (base) bar; therefore, the relative orientation and position of cameras is known with respect to each other. The cameras have the same working distance in this way, and therefore are easily removed from photogrammetry location measurements as a variable [100].

When the load is applied, the pattern is deformed as the object being inspected is deformed. The structural deformation under specified loading conditions is captured and recorded by two DIC cameras. Unique correlation areas, which are called facets, are defined by initial image processing across the whole imaging area, and normally range from 5 to 20 square pixels in size [102–104]. Every consecutive pair of images is tracked with sub-pixel precision from the measurement point located at the centre of each facet. The movements of these facets are tracked by an image correlation algorithm by using mathematical techniques to achieve maximum similarities determined from consecutive photographs [102]. The software of image correlation is essentially designed with the purpose of pattern matching, which can be performed on both curved and flat surfaces [100]. The locations of each facet in three-dimensions can be determined before and after every loading stage while examining in this way, and therefore resulting in three-dimensional displacements, the plain strain tensor, and the three-dimensional shape [103,104]. Data of full-field displacement can be acquired from the measurement facet point tracking in the applied regular target patterns.

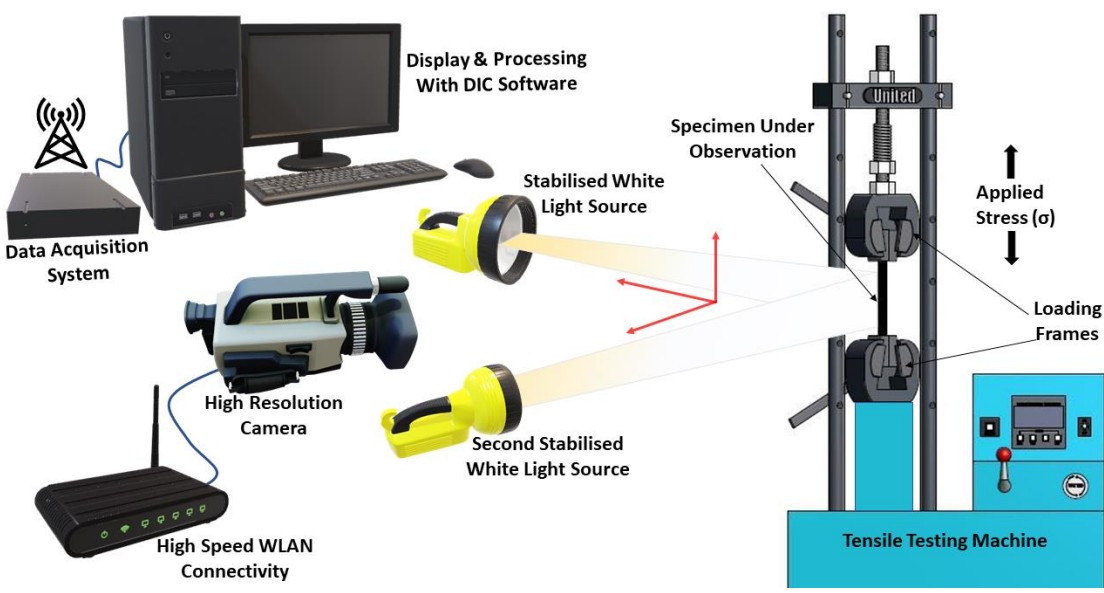

**Figure 10.** Working setup and principle of digital image correlation method-based NDE of composites.

DIC has gained popularity and has numerous commercial applications since it only employs high-speed computers and high-resolution digital cameras, and therefore is economical. The overall precision of DIC increases with the increase of megapixel resolution of the digital camera used, because it makes more measurement points available for comparison [102]. A recent approach of a multicamera measurement system for evaluating the dynamic response from wind turbine blades has shown the capability to detect in-plane displacement as low as 0.2 mm [1]. However, the limitation in DIC is that both the cameras have an optimum total angle of 25°, and these lower angles decrease the triangulation accuracy, therefore reducing the out-of-plane (*z*-axis) displacement accuracy [102]. The wider optimum angle increases the *z*-axis measurement accuracy but can result in a decrease of the overall view field. The DIC method can be modified for use in either static measurements, which are at low frame rates, or dynamic measurements with higher frame capture rates in frames per second (fps). Static measurement provides the chance of conducting long-term condition monitoring and also captures comparatively slow loadings at 1 fps or less [100]. Dynamic measurement is used in the DIC test, in which the rate of data acquisition is even more than 1,000,000 fps [100]. In either case, the data transmission can be performed remotely using a Wireless Local Area Network (WLAN) connection, which means even remote-controlled drones can be employed for this method. Hence, DIC can also be employed for humanly difficult to access regions, depending on the available capability of drones. Therefore, DIC is a very flexible NDE method which can be modified according to the test environment. However, it can only be effectively used for identifying surface defects or changes.

### 3.3. Electromagnetic Spectrum-Based NDE

Electromagnetic testing techniques utilise an electric current, magnetic field, or both to induce a response from a test piece, and the received electromagnetic response is observed to identify and examine defects, fractures, etc. Some of the popular electromagnetic techniques that include eddy current testing, infrared thermography, and FMCW are discussed herein. Other variants that are also used under the electromagnetic spectrum but are not discussed in length herein are: electrical impedance spectroscopy used for measuring the impedance response from CFRP composites [105], broadband dielectric spectroscopy used for damage assessment by measuring the dielectric response of composites [106], and electrical impedance tomography used in filament wound composites for NDE sensing [107].

### 3.3.1. Eddy Current-Based NDE

Eddy current testing utilises an electrical coil through which a magnetic field is generated, and if the sample is a conductive material, a circular electric current is created. This circular current helps to identify the crack existence, surface damage, the difference in sample composition, and even the identification of material variations itself. The working principle is presented in Figure 11. This method falls under electromagnetic testing and is one of the oldest characterisation techniques [108]. In this method, the changing magnetic field is produced by passing an alternating current through the coil. The magnetic field induces an eddy current or circular current if the coil is located near the conducting material, wherein the presence of any defect would modify this generated field. The ability to monitor phase and magnitude changes in the concentrated eddy current over the sample gives this sensing method the ability to detect cracks and corrosion damages, and measure coating thickness, material thickness, and material conductivity to identify the overheating damages, and is also very useful to monitor the heat treatments [109].

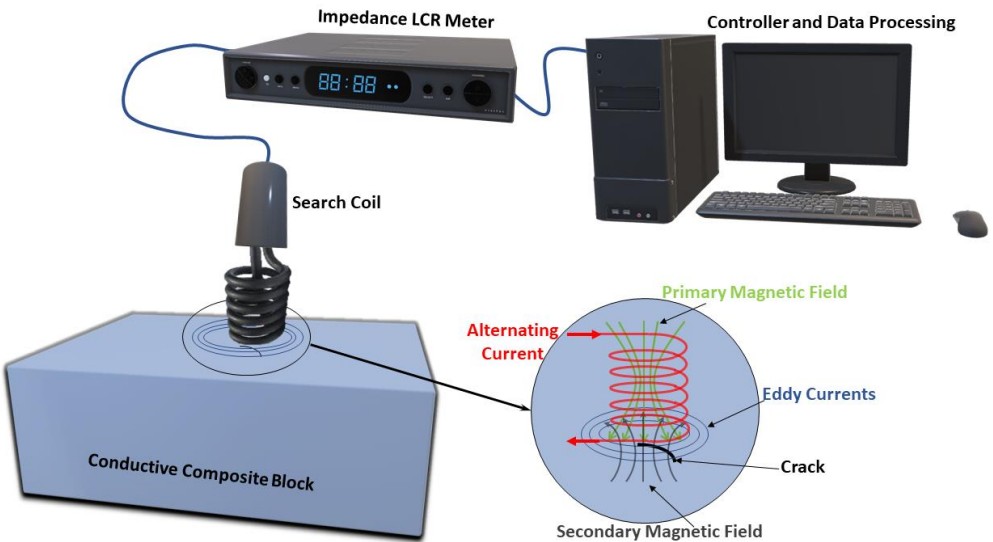

**Figure 11.** The working setup and principle of eddy current testing-based NDE of composites.

The advantage of eddy current testing includes superior sensitivity (compared to methods such as acoustic and ultrasonic testing) to small surface cracks and the other defects which are located over and below the surface layer, since the concentration of the eddy current is quickly recorded in such cases. It can also examine complex parts, in terms of their surface contours and the nominal material preparation time that is required with the capability of portable equipment [109]. However, there are also some limitations, which include: only materials with electrical conductivity can be examined (such as carbon fibre-reinforced polymer (CFRP) composites), rough finishes can obstruct the examination, the surface must be reachable by a probe, requires exceptional inspecting skills and experience by the operator, and its unsuitability for large-area examination [110]. The setup of this technique requires a testing coil, alternating current source, and a suitable display, as shown in Figure 11.

There are also more advanced techniques of eddy current testing, such as eddy current holography. This method characterises the conductive composite structure integrity with different types of discontinuities, for example corrosion and delamination, and surface and close to surface defects. The eddy current holography method is utilised to visualise the delaminated areas in quasi-isotropic composite structures, and delamination generated by impact energies can be efficiently inspected [111]. Recent versions of high-speed non-contact eddy current measurements have made the rapid assessments of unidirectional CFRP structures for delamination defects feasible, at an increased rate of 4 m/s [112,113].

Eddy current testing has faced some further limitations when used for NDE of carbon fibre composite structures [114,115]. There is difficulty in interpreting the measured signals, for example, determination of interlaminar crack delamination. The depth of penetration is minimal for the detection of most surface and subsurface flaws. The method is limited to composite structures which are composed of conductive fibres, for example, carbon fibre, and most of the time requires modifications for lower conductive materials. Furthermore, in industries, the application of eddy current testing is still limited because of the many intrusion aspects, for example, the resulting eddy current is easily influenced by any conductive component in the vicinity. Finally, the lift-off effect is required to be considered, which includes the variation in the mutual inductance between the test sample and excitation coil because of the changes in distance between the test sample, probe, and surface conditions [116].

### 3.3.2. Infrared Thermography-Based NDE

Also known as thermal imaging, thermography testing is a thermal radiation-based technique, which is recorded using the infrared camera and emitted by the surface of the sample. The working principle for infrared thermography is shown schematically in Figure 12. The presence of defects and flaws, such as impact damage or delamination, changes the material thermal behaviour, leading to localised differences in the emitted, transmitted, or reflected infrared emission of the sample, which can be detected by thermography measurements [117,118]. When the defect is deep below the surface (up to 4 mm) in thin components, less heat fluctuation is produced than the heat produced by the defects which are located close to the sample surface. As a rule, the presence of defects in a structure that have a smaller dimension (length, width, or diameter) than their depth is not able to be detected by this testing method. Thermography is popular for detecting impact damage, delamination, cracks, structure debonding, and water ingress in honeycomb structures [27].

Thermography can be operated in either a passive or an active mode. Passive thermography directly measures the surface temperature for evaluation, since the region of interest will exhibit an abnormal hot spot when compared to the surroundings and wherein an abnormal temperature profile indicates a potential problem. Active thermography measures the surface temperature for evaluation after applying some thermal excitation, wherein the defects can be detected by an anomalous heat transfer response evolving after a certain applied excitation time. Passive thermography is normally utilised for materials which are not thermally balanced and possess temperature contrasts with the neighbouring environment, for example this can be used for examination of water ingress after the landing of aircraft because of the considerable temperature difference between the aircraft material and water [119]. However, in active thermography, the material is exposed to thermal energy externally to induce the temperature difference between the required areas by utilising various heat sources or even cold sources [120,121], or both energy sources applied simultaneously on opposite regions [122], or both sources applied subsequently on the same region [123]. Active thermography is the most widely used technique for NDE of composite aerospace parts [4,124], and it can be subdivided into acoustic/ultrasonic-stimulated thermography [125–128], eddy current-stimulated thermography [129], indirect material-based thermography (metal-based, carbon nanotube-based, and shape memory alloy-based) [130–132], and optically stimulated thermography (pulsed/flash, lock-in/amplitude modulated, step-heating, long pulse, frequency modulated, laser-spot, and laser-line type thermography) [126,133–138].

This method has many advantages and disadvantages. One prominent advantage includes the ability to examine large surfaces of the component. Another advantage associated with this method is that it does not require physical coupling with the test component, similar to some of the other inspection types previously discussed. This helps in inspecting the parts in which only one side is available for examination. The drawback associated with this method is the requirement of sensitive and costly instrumentation,

highly skilled examiners required to work on the equipment, and not having defect clarity if it is present deep under the surface [139].

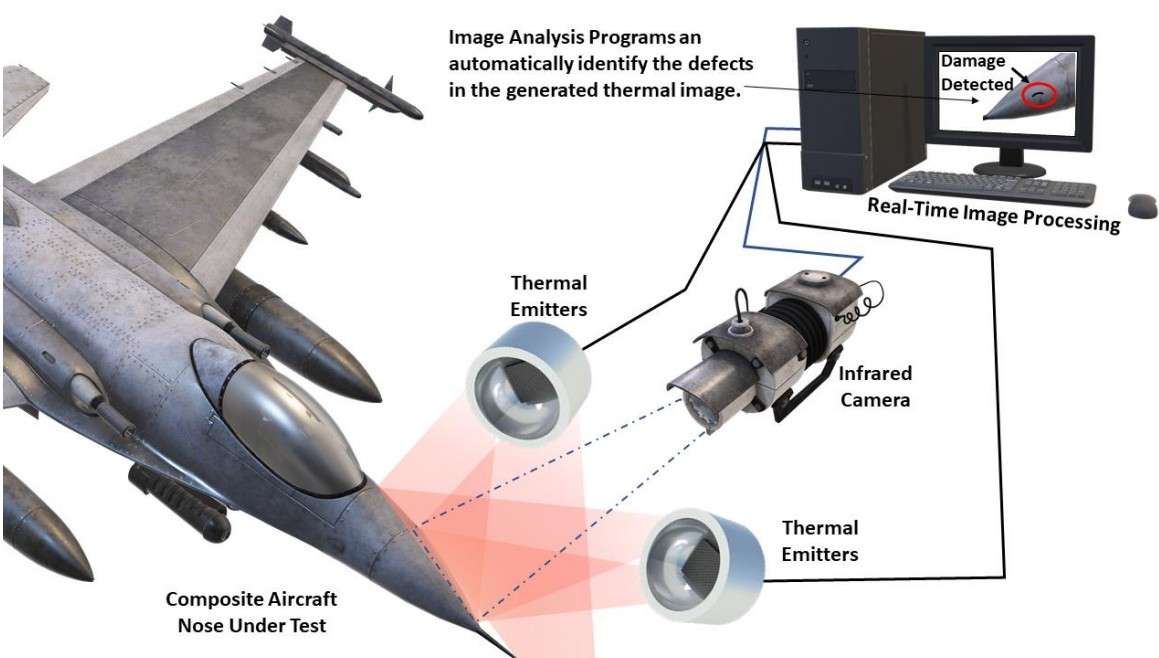

**Figure 12.** The working setup and principle of thermography testing method-based NDE of composites.

### 3.3.3. Frequency-Modulated Continuous Wave-Based NDE

Frequency-modulated continuous wave has been adopted in ultrasonic fields for NDE applications over the last decade [140–142] and has been previously utilised for radar and optic applications [143]. The ultrasonic techniques coupled with FMCW systems have two types with respect to more standard approaches, which are pulse-echo [144] and pulse compression [145–147]. FMCW systems display unique features which have practical advantages in real-world applications. The simplest scenario of FMCW radar sensing, as mentioned in [148] and as shown in Figure 13, consists of a wave generator, transmitting and receiving unit, and processing and display unit.

In this method, the FMCW system consists of a radar horn antenna that acts as a transmitter and receiver. The process of measurement begins with the emitting ultrasound transducer excitation with the periodic chirp signal, reaching to the defined frequency interval in a time achieved by a digital to analog (D/A) converter fed with a proper digital sequence within the signal generator. The emitted ultrasonic signal travels within the exposed medium and reaches the receiving transducer, and acquires information about the medium, such as propagation delay [148].

It was also recently presented for characterisation of single-layer dielectrics [149]. The FMCW transceiver, specifically with regards to industrial applications, can obtain a kilohertz measurement rate at a higher integration level. The sensor bandwidth and the layers' refractive index are used to determine the inherent FMCW radar resolution limit, for instance, the thickness of a few millimetres and below of the FMCW systems can be resolved with the bandwidths of 40 to 90 GHz [150]. The MHz bandwidth of the FMCW radar within 300 MHz to 300 GHz is found useful to study defects in the range of 1000–1 mm respectively [27].

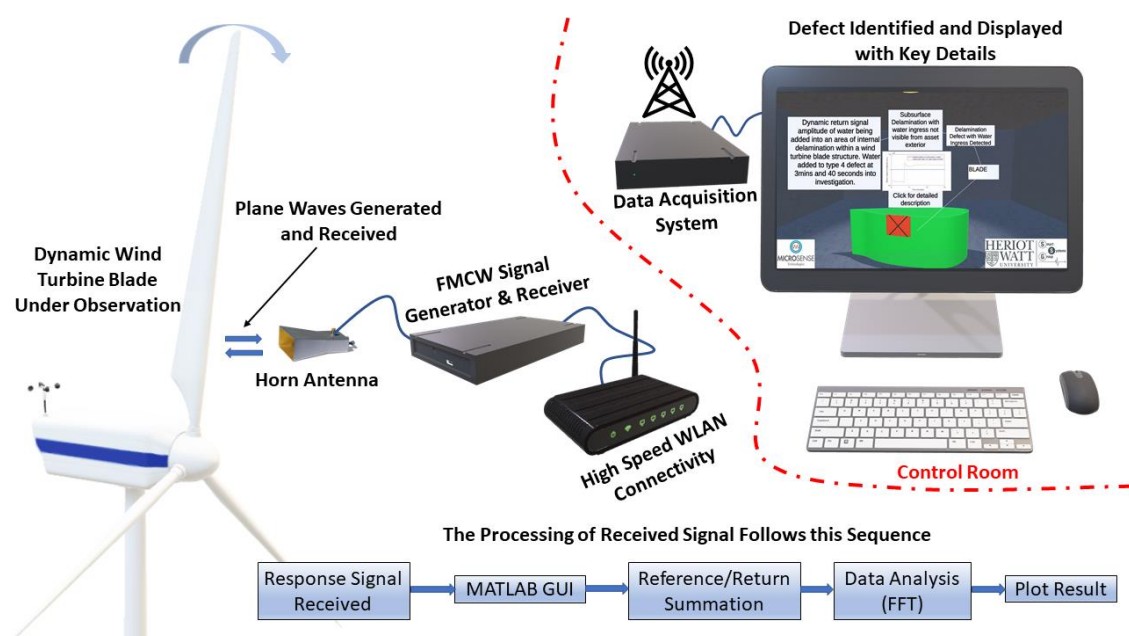

**Figure 13.** Setup of frequency-modulated continuous wave radar system-based method for NDE of composites.

Though FMCW has recently been applied as a stand-alone system for NDE of WT blades for studying the delamination, cracks, and water ingress [35,37], the literature shows that there is a wide area of applications for FMCW-based NDE of composites. Due to its nature of interaction with dielectric materials, it can also be used to identify variable materials present in a composite material [151], and the concept can even be extended for a micron-level nanoparticle agglomeration study, which is a critical aspect in bespoke polymer nanocomposites [152–154]. However, similar to other methods, it also has some limitations, which include a limited depth of penetration against other methods involving ground penetrating radar, X-ray, Gamma, and neutron [155], in addition to spatial resolution, which is limited by the bandwidth and low power, which limits the penetration depth in the target composite [18,35,36,155].

The resolution limitation of MHz FMCW is widely overcome by shifting to a higher bandwidth of the microwave spectrum, which is called continuous wave terahertz imaging, or popularly categorised as the THz NDE method [156,157]. This method is known to easily provide a resolution of up to the sub-millimetre range, with a proven performance of identifying embedded wire of 35 μm in diameter and water ingress [156]. Though the resolution attained herein is higher, as the bandwidth is increased from GHz to THz, there is a corresponding significant reduction in target material penetration.

## 4. Comparison of NDE Sensing Technologies

Composites have advantages when compared to mechanical properties of metals with reduced weight potential, which is the main reason for their increasing interest in different applications. Though there are many challenges encountered in NDE of composites, the NDE techniques discussed in this paper are eddy current testing, ultrasonic testing, acoustic emission, shearography testing, thermography testing, computed tomography, digital image correlation, piezoelectric transducer, and FMCW, with respect to their advantages and limitations, the summary of which is presented in Table 1 below.

**Table 1.** Summary of NDE methods with their advantages and limitations.

| NDE Methods | Types of Defects Identified and Resolution | Advantages | Limitations | References |
|---|---|---|---|---|
| Eddy current-based NDE | • Surface and subsurface defects.<br>• CFRP Delamination.<br>• Linear defects (up to 0.5 mm deep and 5 mm long). | • Examine complex parts.<br>• Nominal material preparation time is required with the capability of portable equipment.<br>• Non-contact type. | • Limited to conductive materials.<br>• Difficult to interpret the measured signals for example differentiation of delamination from interlaminar cracks.<br>• Parallel defects to the test object can go undetected.<br>• Depth of penetration is not very large to detect for most of the surface and subsurface flaws. | [14,38] |
| Shearography-based NDE | • Surface defects—damage, delamination, substrate defect, debonding (bonded laminate defects), and water ingress (limited applicability).<br>• Detect features up to 2–3 mm depth or sometimes even more. | • Less prone to noise than other different kinds of NDE.<br>• Does not require highly skilled operators for the inspection and determination of component usability without long training. | • The inspection of subsurface defects other than delamination is very difficult.<br>• It is necessary to combine this technique to other NDE methods for detailed analysis. | [6,14,27,38] |
| Infrared thermography-based NDE | • Impact damage, delamination, cracks, structure debonding, honeycomb water ingress.<br>• Small surface defects (0.1–0.5 mm) and defects only a few mm (up to 4 mm) under the surface.<br>• Large cracks of up to a few millimetres. | • Can determine the change in the material thermal conductivity by the presence of defects and flaws.<br>• Inspect large surface areas.<br>• Does not require to couple like other inspection types.<br>• Helps in inspecting the parts in which only one side is available for examination. | • Requirement of sensitive and costly instrumentation.<br>• Highly skilled examiners required to work the equipment.<br>• Not having defect clarity if it is present deeper under the surface. | [27,38,158] |
| Computed tomography-based NDE | • Surface and subsurface defects.<br>• Cracks and delamination.<br>• Microscopic failures within the component.<br>• Detectability of up to 0.2 μm with nanoCT scans. | • Determine internal features of structure and to attain digital knowledge on their 3-dimensional geometries.<br>• Modifies the scale from macroscopic to microscopic | • Sample size greatly affects the information obtained by current generation of computational tomography systems.<br>• Field of view limitation.<br>• Limited for in situ monitoring.<br>• Attenuation contrast. | [6,14] |
| Electrostatic transducer-based ultrasonic NDE | • Surface and subsurface defects.<br>• Cracking and delamination.<br>• Wall and defect thickness measurement.<br>• Discontinuities or damages within 1–5 mm. | • Flaw detecting capabilities with good resolution and speed of scan.<br>• Portable and can be used in the field.<br>• Helpful to test in the assembly line in which same design parts are frequently tested. | • Only contact type and may need a couplant for maintaining contact.<br>• Not suitable for in-depth flaws' characterisation due to confusing signals.<br>• Shadow effect while testing aerospace composites.<br>• Difficult to set up as both transmitter and receiver are to be handled accurately.<br>• Requires accurate part scanning and mounting skills assure precise examination of test specimen.<br>• Difficult to use in thin materials.<br>• Relatively smooth surface is needed to couple transducer. | [6,14,27,38,51] |

Table 1. *Cont.*

| NDE Methods | Types of Defects Identified and Resolution | Advantages | Limitations | References |
|---|---|---|---|---|
| Piezoelectric transducer-based ultrasonic NDE | • Surface and subsurface defects.<br>• Cracking and delamination.<br>• Wall and Defect thickness measurement.<br>• Critical defects of relatively large (up to 25 mm) sizes. | • Provide flexibility and wide bandwidth.<br>• High flexibility of piezocomposites in inspecting curved pipelines or steam pipes monitored as pipe curvatures.<br>• Useful in inspection of non-porous and homogeneous materials. | • Limited applications at higher temperature and power because of their naturally high thermal expansion of polymer fillers, low thermal conductivity, and low mechanical quality factors. | [27,60] |
| Acoustic emission-based NDE | • Delamination crack growth, matrix cracking, fibre misalignment, and fibre breakage.<br>• Surface and subsurface defects.<br>• Defects from sub-millimetre to up to 5 mm. | • Quick and overall testing by means of multiple sensors, high sensitivity, and sensor mountings for process control.<br>• No requirement of disassembling and sample cleaning.<br>• Helpful in detecting various defect types resulted from fatigue loading. | • Only contact type testing and may need a couplant for maintaining contact.<br>• Requires high skill for correlating acoustic emission data to explicit damage mechanism type. | [6,14,38] |
| Digital image correlation-based NDE | • Surface defects—impurities, inclusions, damages, and substrates.<br>• Detect in-plane displacement as low as 0.2 mm. | • Non-contact.<br>• Economical.<br>• High-resolution digital cameras and high-speed computers.<br>• Wider optimum angle increases the $z$-axis measurements' accuracy. | • Lower optimum angles decrease the triangulation accuracy, and therefore reduce the out-of-plane ($z$-axis) displacement accuracy.<br>• Wider optimum angle results in the decrease of overall view field. | [1,100] |
| Frequency-modulated continuous wave-based NDE | • Surface and subsurface defects.<br>• Delamination.<br>• Larger size defects and deep penetration.<br>• Microscopic failures on surface and within the component.<br>• Debonding and water ingress.<br>• Foreign material detection.<br>• Microwave bandwidth (300 MHz–300 GHz) can detect 1000–1 mm size defect.<br>• Terahertz bandwidth (300 GHz–3 THz) can detect defects from 1 mm to 35 μm. | • Non-contact.<br>• Resilient in opaque conditions, even in harsh and humanly inaccessible environments or locations.<br>• Capable of surface and subsurface defects.<br>• Lightweight and portable.<br>• Biologically safe at low power. | • Spatial resolution limited by bandwidth.<br>• Low power limits penetration.<br>• Cannot be used in underwater testing. | [18,27,35,36,149,155,156,159] |

## 5. Future of NDE of Structural Composites

The market for NDE of composites is growing steadily, with increasing involvement in many application areas, as summarised earlier with a few select examples in Figure 1. The global market for composite testing is forecasted to be worth USD 3.34 billion by 2027, and the region-wise breakdown is presented in Figure 14. The demand for NDE of structural composites is ever-increasing, and hence a major research focus has been directed towards its development. With the development of advanced disruptive technologies, there has been a focus on improving the existing methods as well. There are many challenges related to NDE techniques in the composite materials field, with the most impending being the analysis and interpretation of the huge amount of data accumulated while testing. The solution to these challenges would be the execution of artificial intelligence and machine learning in pattern recognition processes and in data processing in NDE methods [40], otherwise these techniques are generally time-consuming and require extremely skilled operators. These can also be minimised by artificial algorithms or network coding to allow automatic inspection and identification of flaws and defects, reducing human error [160].

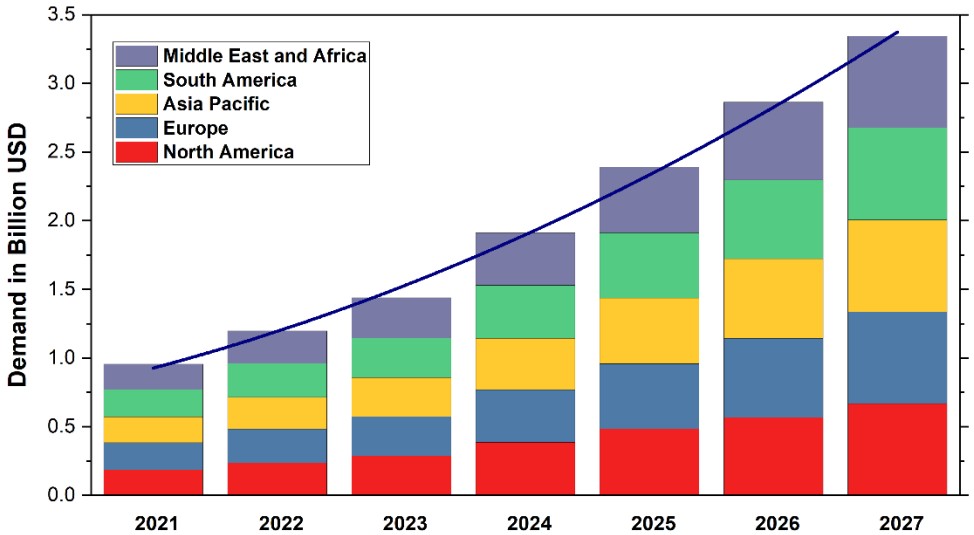

**Figure 14.** Region-wise market demand forecast for the NDE of composites for the years 2021–2027, with the trend line overlayed [161].

Research on signal processing and statistical analysis approaches to resolve problems in NDE has been the major focus of scientists and engineers for decades, especially for the NDE signals' interpretation for flaw detection and characterisation. For example, clustering (to identify natural clusters in collected signals) has been observed to have a number of applications in acoustic emission signals' analysis [162]. Clusters in signals can be utilised to separate out different types of acoustic emission signals. These signals are separated and correlated with defects such as fibre breakage, matrix cracking, and interface failure [163]. Other machine learning approaches include matrix decomposition and neural networks. Matrix decomposition techniques can be used in guided-wave PHM for the separation of damage events and their changes over longer time periods, of over 10 to 1000 to 100,000 s of guided-wave measurements, and can be used to obtain information in the presence of noise and temperature variation environments from the dataset [164]. Neural networks can be used in NDE applications, such as in ultrasonic signals' classification for crack detection [165]. This method can also be used for defect localisation, material properties' characterisation, and damage characterisation. The future of composites' NDE is headed in the direction of data-driven approaches, opening up newer dimensions of NDE efficiency by involving methods such as deep learning, transfer learning, and physics-informed machine learning, with multiple articles already published [166,167].

The future of NDE includes investigations into new sensing modalities for composites, such as extending the preliminary work of Tang et al. on FMCW radar sensing for wind turbine blades [168]. This technology has also been used for other material characterisation challenges, using both static and dynamic loads, for example, to identify damages for surface and subsurface structural analysis and act as a discriminator for the presence of porosity elements and to identify both frozen phase and liquid phase of fluid (water) ingression [37,155]. Using this sensing mechanism, the further deterioration of the wind turbine blade can be detected and prevented, increasing the operational efficiency of the blade. FMCW radar sensing is an advanced NDE technique for the real-time defect detection. The FMCW radar sensor has advantages of being a non-contact and non-destructive method which is unaffected by smoke, mist, and fog. FMCW sensing technology has versatility to improve the current state-of-the-art in defect detection approaches, such as visual and thermal inspections, which can be improved by coupling novel digital analysis and digital twin systems with FMCW systems in the future [35].

Digital analysis by the data-driven approach is an advanced and rapidly evolving method in sensing technologies. These digital tools are becoming important to support the mixed integration of information and data from distributed monitoring systems because of increasing complexities in systems and dependencies across system networks, such as NDE techniques. Digital twin (DT) is a digital representation of devices and a physical system with respect to their lifecycle and environment, similar to a mirror of a physical object that creates a relation between the physical and virtual objects. It is a direct and spontaneous method which can improve human–object interaction with less specialist knowledge requirements from end users. It is expected that the DT framework will be further evaluated in other real-world scenarios, such as robotic platforms, for proving that operators can fully interact with their physical assets such as in NDE technology by the internet and visualised virtual workspaces [169].

The implementation of such advanced technologies will provide high potential for NDE methods, for example NDE techniques based on terahertz have already confirmed their benefits with regards to penetration depth, and techniques such as free electron lasers and modern spallation sources have shown potential. Further, the next-generation synchrotron X-ray and neutron also offers added novel capabilities [6,14,38,160].

## 6. Conclusions

Application of the NDE methods for composite materials has become in demand and critical because of their use in critical safety applications. This review presented a comprehensive overview of the currently available NDE techniques and was categorised according to the method utilised. These NDE techniques have been explained in detail by describing their working principle, advantages, and limitations. It was observed that most of the NDE inspection involves on-site examination and inspection of comparatively large and complex geometry structures, which limit the application of NDE methods. It also requires high competence levels and highly skilled inspectors. The market trends of the technologies based on various market reports was also discussed in the review. It was demonstrated that the market of such technologies is expected to grow at 6–7% CAGR by 2025. Going forward, it is anticipated that the future NDE methods will involve implementation of artificial intelligence and machine learning to minimise human error. The next-generation NDE includes an advanced application of FMCW radar sensing as a major contender and was presented with an example application for wind turbine blades, to identify damages by surface and subsurface structural analysis. A comparison of the discussed NDE methods was also presented as a summary guide for the benefit of the reader. The comparison suggests that FMCW sensing technology has the versatility to enhance the current state-of-the-art in-service defect detection approaches. The future trend indicates that the NDE methods' efficiency can further be improved by coupling novel data-driven digital analysis and digital twin system deployments.

**Author Contributions:** Conceptualization, R.G.; writing—original draft preparation, R.G.; writing—review and editing, R.G., D.M., J.B., S.H., W.T., K.P., L.B., D.G.B. and D.F.; graphics and visualization, R.G.; funding acquisition, D.G.B., D.F. and L.B. All authors have read and agreed to the published version of the manuscript.

**Funding:** The authors are grateful for the financial support provided for this research by Innovate UK, Heriot-Watt University and MacTaggart, Scott & Co Ltd., through KTP Grant Reference No. 11746.

**Conflicts of Interest:** The authors declare no conflict of interest.

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
