# Peer review of "A Review of Sensing Technologies for Non-Destructive Evaluation of Structural Composite Materials"

_jcs, doi:10.3390/jcs5120319_

Round 1
Reviewer 1 Report
This paper presents a review on the sensing techniques employed in non-destructive evaluation (NDE) of composite structures. Various NDE methods have been introduced in detail in terms of the measuring principles, setups, advantages, limitations, as well as the current usage levels for structural composites. The overall structure of the manuscript is sound, containing sufficient information and concluded insightfully for future research. Some suggestions are summarized below for further improvements:
(1) Section 2 Market Trend for NDE of Composites, academic publications usually reflect the progress in advances and novel NDE methods applied in structural composites, which not necessarily connected to the market trend for NDE of composites. A review or search specifically towards their industrial application trends would be helpful.
(2) Section 3, it is suggested to report and include the most advances in detection resolution of each NDE method applied in composite materials.
(3) Line 351, the figure number is incorrectly quoted here.
(4) Line 615, the abbreviation “WT” is not defined in the text.
(5) Line 631 it should be “computed tomography”, other than “computed thermography” in the text.
(6) Table1, again it should be “computed tomography”.
(7) One single bracket is detected in Line 666.
Author Response
Reply in attached Document.

Reviewer 2 Report
The manuscript - "A Review of Sensing Technologies for Non-Destructive Evaluation of Structural Composite Materials" presents an interesting overview of the NDE technology currently available for the inspection of composites. This is an interesting paper but needs some revisions before being acceptable for publication.
The authors state that the current research trends show the emergence of non-contact NDE techniques. Non-contact NDE is definitely a trend but not only, there are 3 main trends that current research is focusing: non-contact, techniques that produce an image, which allows overlapping the result with the inspected part for easier results interpretation, and techniques that can be automatized.
I believe some of the techniques should include some more relevant works and there are some other techniques that are not even mentioned and should. I recommend the analysis of a few works:
Thermoelastic stress analysis is not mentioned.
10.1016/j.ndteint.2021.102526
10.1007/s10921-019-0564-y
10.1016/j.compstruct.2018.10.047
Vibrothermography is also overlooked.
10.1007/978-3-030-44522-5_8
10.1016/j.compstruct.2018.06.105
10.1016/j.compstruct.2019.02.003
Figure 4 mentions the THz gap which is no longer a gap. It is a very emerging technique in the NDE topic, and it is only mentioned in the last paragraph of the manuscript with no citations. This technique should be further explored.
10.1016/j.ndteint.2021.102473
10.1016/j.ndteint.2020.102383
10.1016/j.measurement.2020.108904
I do not understand why the US inspection was divided in section 3.1.1 and section 3.1.3 since they actually cover the same technique. In section 3.1.1 the authors state that the ultrasound testing method comprises a transducer tool. This transducer tool consists in what? Is it not a piezoelectric?
Would make more sense to have a section for conventional US (with contact through a couplant) and air coupled or non-contact US (which the authors called ““through transmission”) much more suitable for polymeric matrix composites due to the low frequency, 50 to 400 kHz.
In line 202 the authors state that the transducer tool comprises a transmitter and a receiver unit. Can’t the transducer tool have both functions? Figure 5 displays the pulse eco US results but then shows a transmitter and receiver probe. Figure 7 makes more sense. When in pulse eco US one probe with both functions is usually employed. When in transmission mode with a transmitter and a receiver usually the probes are on opposite sides of the component under test (particularly in air coupled US).
In the eddy currents technique, a few very relevant works are missing which cover the high-speed inspection and its performance on the detection of fibre breaks and delaminations of unidirectional CFRPs.
In the active thermography the external thermal energy is usually heat but it is not limited to heat, a cold source has been used (10.1080/17686733.2019.1625243 and 10.1016/j.ndteint.2018.11.012), both energy sources at the same time in opposite surfaces (10.1016/j.ndteint.2021.102566) or one after the other on the same surface (10.1016/j.infrared.2021.103860).
In line 104 “to read the stress and stress at the location” one of the stress’s is probably meant to be strain?
In line 507 “Eddy current testing utilises an electrical coil through which a magnetic field is generated and if the sample is a metal, a circular electric current is created” Eddy currents can form in any conductive material, it is not limited to metals (almost).
In line 513 “The magnetic field induces an eddy current or circular current if the coil is located near the conducting material having a crack” The magnetic field induces eddy currents when located near conducting materials regardless of having cracks or not.
In line 666 “10s to 1000s to 100,000s” numbers and units should always be spaced.
Author Response
Reply in attached document.

Reviewer 3 Report
It is interesting and welcoming that the authors are submitting a Review article on NDE of Composite Materials for the Journal of Composites. The article is very well written and very well structured.
However, there is a major concern that the review article lacks. Review Articles are typically written as a complete collaboration to identify all previous literature, and work done in the field. It starts on each topic with an introduction to the topic and cites relevant papers and research being done. Further, it lists the demerits and identifies the gap, and proposes the solution as future work.
In this present article being considered, The authors have done a very good job explaining the content, listing demerits. However, the Gaps are not identified and the need for this article is not explained.
There are so many similar articles and books on NDE, SHM of composite materials, and there is a lot published in the recent 2020,2021 years. In that scenario, the need for this review article or how this article is different from others is not explained.
According to the reviewer's understanding, this article's main aim seems to appear that, the authors wanted to write a review article, and also showcase their work past and future being done, by citing and exhibiting their work.
For more explanation, a review article needs to be focused on a particular topic - as in if it's about data-driven methods or AI, it can be on AI-based NDE techniques, or if it's about a particular methodology it can list those methods, or at least focused on a timeline, i.e list only past 5 years work in the field, stating that there is no review on the past 5 years. Hence, the need for this article and how it's different from others need to be explained.
Minor Comments.
1. Cite previous review articles and state how this article is different from them - doi.org/10.1016/j.prostr.2016.02.008 ; doi.org/10.1016/j.cja.2019.09.017 ; doi.org/10.1016/j.renene.2020.07.145 ; doi.org/10.1088/1361-665X/ac099f ; doi.org/10.1177/1687814020913761 ;
2. Briefly State the difference between NDE, SHM, and Prognostics
3. Line 104 - "Stress and Stress at the location " - is this a typo or do you refer to global stress and local stress?
4. Among Electromagnetic Spectrum Based NDE, there is another field that relates to Electrochemical Impedance Spectroscopy or Broadband Dielectric Spectroscopy, or Electrical Impedance Tomography - these techniques are used for NDE of composites and are widely known. Please refer to them and cite a few articles about them; Referring to other review articles will give a better understanding of all the techniques that need to be considered in this article
5. Lines 335 to Lines 350 are Plagiarized from www.ndt.net ; Please rephrase or rewrite this section 3.2.1
6. About 20 of the team's work is being cited. Few are cited with no major significance, but because the article being cited has some small content in its introduction section relating to this content. such articles can be removed.
Author Response
Reply in attached document.

Round 2
Reviewer 2 Report
The revised manuscript is recommended for publication.
Reviewer 3 Report
Thank you for making all the corrections as suggested and for providing point-to-point responses for each of the comments! Kudos on this work and looking forward to the future contributions in this field